# COVID-19 Prevention Guidance and the Incidence of Febrile Neutropenia in Patients with Breast Cancer Receiving TAC Chemotherapy with Prophylactic Pegfilgrastim

**DOI:** 10.3390/jcm11237053

**Published:** 2022-11-29

**Authors:** Hongki Gwak, Seung-Taek Lim, Ye-Won Jeon, Hyung Soon Park, Seong Hwan Kim, Young-Jin Suh

**Affiliations:** 1Division of Breast and Thyroid Surgical Oncology, Department of Surgery, Hwahong Hospital, Suwon 16630, Republic of Korea; 2Division of Breast and Thyroid Surgical Oncology, Department of Surgery, St. Vincent’s Hospital, College of Medicine, The Catholic University of Korea, Seoul 16247, Republic of Korea; 3Division of Medical Oncology, Department of Internal Medicine, St. Vincent’s Hospital, College of Medicine, The Catholic University of Korea, Seoul 16247, Republic of Korea; 4Department of Plastic and Reconstructive Surgery, Kangnam Sacred Heart Hospital, Hallym University College of Medicine, Seoul 07441, Republic of Korea

**Keywords:** COVID-19, infection prevention, febrile neutropenia, breast cancer

## Abstract

Chemotherapy-induced febrile neutropenia (FN) is a medical emergency that causes severe adverse effects and death. Respiratory infections are one of the main causes of fever in patients with FN. We studied whether infection prevention and control (IPC) guidance for coronavirus 2019 disease reduced the incidence of FN. We reviewed female patients with breast cancer treated with adjuvant docetaxel, doxorubicin, and cyclophosphamide with prophylactic pegfilgrastim between 2019 and 2021. IPC guidance was implemented in April 2020. There was no difference in the incidence of chemotherapy-induced neutropenia between patients with and without IPC. In patients with IPC, the incidence of FN (9.5%) was lower than that of patients without IPC (27.9%). The hospitalization duration (0.7 ± 1.5 days) and total hospital cost (279.6 ± 42.6 USD) of the IPC group were significantly lower than that of the non-IPC group (2.0 ± 3.8 days and 364.7 ± 271.6 USD, respectively). IPC guidance should be implemented to prevent FN in high-risk patients with breast cancer.

## 1. Introduction

Breast cancer is a systemic disease, and many patients require chemotherapy in addition to oncological surgery and radiotherapy [1]. Dose-dense anthracycline or taxane- and docetaxel-based regimens are often used to treat breast cancer. According to international guidelines, the docetaxel, doxorubicin, and cyclophosphamide (TAC) regimen carries a high risk (>20%) of chemotherapy-induced neutropenia (CIN) and its complications [2,3].

Patients with CIN may progress to febrile neutropenia (FN), a serious medical problem. TAC causes significantly more hematological adverse effects than other breast cancer chemotherapy regimens [4]. The use of prophylactic G-CSF (granulocyte colony-stimulating factor) improves the prognosis of patients receiving the TAC regimen by lowering the incidence and duration of FN [5]. The National Comprehensive Cancer Network (NCCN), American Society of Clinical Oncology, and European Organization for the Research and Treatment of Cancer guidelines recommend the routine use of primary G-CSF prophylaxis for high-risk cases (risk of FN > 20%) based on the findings of several randomized controlled trials [6,7,8]. The use of prophylactic ciprofloxacin can also lower the incidence of FN [9]. Despite these efforts, the rates of CIN and FN remain high, at 74–91.4% and 7–21.4%, respectively [9,10,11,12]. 

One of the main causes of neutropenic fever is a respiratory infection, which is associated with a high mortality rate. Most respiratory infections in patients with FN are bacterial or fungal, and can be controlled to some extent by drugs. However, viruses are also a cause of fever, and one study detected viruses in 41% of patients [13,14]. After the COVID-19 outbreak, the World Health Organization and Centers for Disease Control promoted mask-wearing and physical distancing. This prevention guidance has proven effective against the transmission of respiratory diseases [15,16,17]. We investigated whether coronavirus 2019 disease (COVID-19) prevention and control (IPC) guidance helped reduce the risk of FN by decreasing respiratory infections in patients with breast cancer receiving TAC chemotherapy.

## 2. Materials and Methods

### 2.1. Patient Selection

This was a single-center retrospective observational study. The study protocol was approved by the Institutional Review Board of the Catholic University of Korea (VC22RASI0019). The electronic medical records of female patients with breast cancer who received adjuvant TAC chemotherapy with prophylactic pegfilgrastim from January 2019 to December 2021 at St. Vincent’s Hospital of the Catholic University of Korea were reviewed. Patients were excluded if they were pregnant or taking immunosuppressive drugs; had previously received other chemotherapy; had abnormal hematopoietic, liver, renal, or cardiac function before chemotherapy; or had an Eastern Cooperative Oncology Group (ECOG) score of ≥2.

### 2.2. Data Collection

The variables extracted from the database included patient demographics, ECOG performance status, cancer characteristics (histological type and grade, stage, estrogen receptor (ER) and progesterone receptor (PR) statuses as determined by enzyme immunoassays, human epidermal growth factor receptor 2 (HER2), surgery type, radiation therapy), laboratory results, and details about infectious diseases. Immunohistochemistry (IHC), fluorescence in situ hybridization (FISH), or silver in situ hybridization (SISH) was used to evaluate HER2 status. Samples with an IHC score of 0 or +1 or an IHC score of +2 and negative FISH/SISH were considered negative for HER2 overexpression. The data were collected at the start of chemotherapy and 2 months after that.

### 2.3. Treatment

The chemotherapy consisted of doxorubicin (50 mg/m^2^), cyclophosphamide (500 mg/m^2^), and docetaxel (75 mg/m^2^), administered on day 1 and every 3 weeks thereafter. Pegfilgrastim 0.6 mL was administered subcutaneously 24–48 h after the chemotherapy. Laboratory tests, including complete blood count (CBC) with differential counts and biochemical assays, were performed before each chemotherapy cycle and on day six. After the chemotherapy, the CBC was evaluated from day 6 until the absolute neutrophil count returned to 1000/mm^3^. All patients with grade 4 CIN received prophylactic antibiotic therapy comprising 2 g intravenous cefoperazone and 200 mg tobramycin sulfate once daily unless their use was contraindicated.

### 2.4. Definitions

ECOG performance status describes a patient’s level of function and self-care capability [18]. It ranges from 0 to 5, with 0 indicating an active person able to carry out all pre-disease activities without restriction and 5 indicating death. The performance status of each patient was evaluated before the first chemotherapy cycle. Adverse events of interest were based on the Common Terminology Criteria for Adverse Events (ver. 6.0).

FN was defined as neutropenia (<500 neutrophils/μL or <1000 neutrophils/μL for over 48 h) with a febrile event (oral temperature ≥ 38.3 °C or ≥38.0 °C for over 1 h) observed by medical staff. Dose reduction was defined as a reduction in the delivered dose of any agent relative to standard values. The total hospital cost was the sum of all medical expenses paid to the hospital during the data collection period. Outpatient hospital visits, hospitalization and chemotherapy expenses, and all devices and drugs used for medical purposes were included in the total hospital cost.

The IPC guidance to prevent COVID-19 infection includes good personal hygiene, handwashing, cleaning high-touch surfaces such as mobile phones regularly or as needed, covering the mouth when coughing and sneezing, and wearing masks. Social distancing involves staying at least 2 m away from others, avoiding poorly ventilated spaces and crowds, and following work and quarantine restrictions [19]. There are three levels of social distancing, implemented according to the number of patients diagnosed with COVID-19 [15]. IPC guidance has been implemented since April 2020.

### 2.5. Statistical Analysis

A chi-square test was used to compare the categorical variables between the IPC and non-IPC groups. Standardized residuals were analyzed for variables with more than two categories. A Student’s *t*-test was used to analyze the continuous variables. To identify the independent risk factors for FN, multivariate logistic regression including variables significant in the univariate analyses was performed. All statistical analyses were performed using R software (ver. 4.0.2, R Core Team, 2013. Vienna, Austria).

## 3. Results

Of the 85 patients who received chemotherapy, 43 (50.6%) followed the IPC guidance. The mean age was 53.7 and 52.3 years in the IPC and non-IPC groups, respectively (*p =* 0.452). Table 1 summarizes the characteristics of the two patient groups. All included patients had an ECOG score of 0, and none had COVID-19 infection during the follow-up period. The age, body mass index (BMI), T stage, N stage, histologic grade, HER2 positivity, and comorbidities were not significantly different between the two groups. Significantly more patients in the non-IPC group were ER-positive (72.7% vs. 52.4%).

Grade 4 CIN occurred in 93.0% (40/43) of patients in the non-IPC group and 90.5% (38/42) of patients in the IPC group (*p* = 0.713). FN occurred less frequently in the IPC group (9.5% vs. 27.9%; *p* = 0.050); however, the difference was not statistically significant. The duration of hospitalization, hospitalization costs, and total hospital costs were significantly lower in the IPC than in the non-IPC group. There were no significant differences between the groups regarding adverse effects, including anemia, thrombocytopenia, liver or renal function abnormalities, and transfusion. There were also no group differences in dose reduction or delay in chemotherapy (Table 2). Of the 14 patients who developed FN, two (12.5%) died of sepsis during the observation period. All deaths occurred in the non-IPC group, but there was no significant difference compared to the IPC group.

The factors that differed significantly between the groups in the univariate logistic regression analysis of FN development were IPC and comorbidities. Age, menopause, BMI, histologic type, T & N stage, and ER and PR statuses had no significant effect on the incidence of FN (Table 3). A multivariate logistic regression analysis revealed that both factors were independently associated with an increased likelihood of developing FN (odds ratio/95% confidence interval: 4.668/1.238–17.602 and 5.554/1.294–23.843, respectively) (Table 4).

## 4. Discussion 

The three chemotherapy regimens most frequently used in patients with breast cancer and axillary lymph node metastasis are AC-T (doxorubicin, cyclophosphamide, and docetaxel), AT (doxorubicin, paclitaxel), and TAC. According to the NCCN guidelines, all three regimens pose a high risk of neutropenia and other complications (>20%) [3]. Our study focused on the TAC regimen, which had a high incidence of FN and was most likely to show the prophylactic effect of IPC measures. Six-cycle TAC has the advantage of similar efficacy, but with a shorter treatment period than eight-cycle AC followed by docetaxel [20]. However, the TAC regimen has a very high rate of FN, with CIN occurring in 100% of patients receiving TAC chemotherapy, and FN in 42.5–63.4% [12,21]. In our study, 97.6% and 18.8% of patients developed CIN and FN, respectively.

Chemotherapy-induced FN in patients with breast cancer usually improves with a brief hospitalization and oral antibiotics. However, in some patients, FN can be a severe adverse effect that requires treatment with intravenous broad-spectrum antibiotics, delays the chemotherapy schedule, reduces the relative dose intensity, and increases the hospitalization period, which increases healthcare costs [5,22]. In addition, FN can cause life-threatening infections, with fatality rates of 5–11% [22]. Our study’s mortality rate of FN was 12.5% (2/14), which was similar to that previously reported. All deaths occurred in the non-IPC group, but there was no significant difference compared to the IPC group. There is no proven method other than G-CSF to reduce the risk of FN. This study is the first to find that IPC can be added to pegfilgrastim to avoid serious complications, FN, and reduce hospital costs. 

In our study, the non-IPC group showed higher ER positivity rates. However, ER expression is known to not affect the occurrence of FN [23]. This study found that ER expression is not a risk factor for FN. Risk factors for FN include age, performance status, sex, comorbidities, laboratory abnormalities, BMI, chemotherapy regimen, neutropenia prophylaxis, tumor type, disease progression, and genetic risk factors [17]. In this study, these factors were controlled or analyzed, and only IPC and comorbidities were independently associated with the development of FN. The incidence of FN was 9.5% in the IPC group, which was lower than the 27.9% in the non-IPC group; however, the difference was not significant (*p* = 0.050). This difference can be verified through future studies with more patients. 

All patients in this study had an ECOG score of 0 and received primary prophylactic pegfilgrastim to prevent FN and its complications and prophylactic antibiotics when grade 4 NIC developed. With these pre-emptive measures, IPC reduced the incidence of fever in the CIN patients included in our study and significantly reduced the hospitalization duration and costs. Mask-wearing and physical distancing have the advantage of not having the potential risk of various pharmaceutical preventive methods and are cheap. Considering these advantages, IPC can be used in chemotherapy patients for whom pegfilgrastim is not recommended.

Respiratory infections are one of the leading causes of fever in patients with CIN [13]. To prevent the spread of COVID-19, protective measures such as mask-wearing and physical distancing have been implemented. In this study, IPC guidance was found to reduce the probability of developing a fever. It is speculated that the main reason is that it reduced the most common cause, respiratory viral infection. However, this study did not analyze respiratory bacteria or virus test results; therefore, further research is needed to prove this speculation. 

This study had some limitations, including its retrospective design. However, it was conducted in a state that controlled the COVID-19 transmission and should have higher reliability than other retrospective studies. Second, the number of patients was insufficient for this single-center study. The TAC regimen had a high incidence of FN, and the preventive effect of the IPC guidance was better than expected. Significant results were obtained despite the small study population. Third, the pandemic situation may have influenced the patients’ treatment policy. Over the 3-year study period, the non-IPC and IPC groups accounted for 15 and 21 months, respectively. During the COVID-19 pandemic, people’s social activities decreased significantly. Patients may have hesitated to visit the hospital even when symptoms occurred, and doctors may have reduced the number of active treatments or follow-ups. These factors may have reduced the number of days in the hospital or the cost of hospitalization. However, by the end of 2021, the cumulative number of patients infected with COVID-19 was approximately 440,000, accounting for only 0.8% of the total population of South Korea, and all patients included in this study were free of infection. Because the number of patients with COVID-19 infection in South Korea during this period was less, hospitalization was not prevented. All patients were treated according to the existing protocol without measures to reduce the number of hospital visits. Due to the limitations of this study, it is difficult to conclude that IPC prevents neutropenic fever. However, it is meaningful that this study dealt with the relationship between IPC and FN, and conclusions can be drawn through additional pragmatic clinical trials.

## 5. Conclusions

IPC measures reduce the hospitalization duration and costs in patients with breast cancer receiving TAC chemotherapy. IPC guidance and comorbidities are independent factors for the development of FN. 

## Figures and Tables

**Table 1 jcm-11-07053-t001:** Patient demographics and tumor characteristics.

	COVID-19 Prevention and Control		
	No (*n* = 43)	Yes (*n* = 42)	*p*-Value
Age (years)	52.3 ± 8.3	53.7 ± 8.6	0.452
BMI (kg/m^2^)	24.8 ± 3.6	24.7 ± 3.7	0.924
BSA (m^2^)	1.6 ± 0.1	1.6 ± 0.1	0.962
Age (years)			
<60	33 (76.7)	28 (66.7)	0.342
≥60	10 (23.3)	14 (33.3)	
Type of surgery			
BCS	39 (90.7)	41 (97.6)	0.360
Mastectomy	4 (9.3)	1 (2.4)	
T stage			
I	26 (60.5)	16 (38.1)	0.074
II	15 (34.9)	25 (59.5)	
III, IV	2 (4.7)	1 (2.4)	
N stage			
0	7 (16.3)	9 (21.4)	0.257
I	27 (62.8)	19 (45.2)	
II, III	9 (20.9)	14 (33.3)	
Histological type			
IDC	37 (86.0)	34 (81.0)	0.531
ILC	2 (4.7)	1 (2.4)	
Others	4 (9.3)	7 (16.7)	
Histologic grade			
G1, G2	28 (65.1)	17 (42.9)	0.051
G3	15 (34.9)	24 (57.1)	
Estrogen receptor			
Positive	33 (76.7)	22 (52.4)	0.024
Negative	10 (23.3)	20 (47.6)	
Progesterone receptor			
Positive	18 (41.9)	12 (28.6)	0.258
Negative	25 (58.1)	30 (71.4)	
HER2			
Positive	8 (18.6)	11 (26.2)	0.444
Negative	35 (81.4)	31 (73.8)	
Comorbidity			
No	38 (88.4)	35 (83.3)	
HTN	1 (2.3)	6 (14.3)	
DM	2 (4.7)	0 (0.0)	
HTN+DM	2 (4.7)	1 (2.4)	0.111
Smoking			
Yes	0 (0)	1 (2.4)	0.494
No	43 (100)	41 (97.6)	

Data are expressed as *n* (%). BMI, body mass index; BSA, body surface area; DM, diabetes mellitus; HER2, human epidermal growth factor receptor 2; HTN, hypertension; IDC, infiltrating ductal carcinoma; ILC, infiltrating lobular carcinoma.

**Table 2 jcm-11-07053-t002:** Incidence of chemotherapy-related adverse events and hospital costs. Data expressed as numbers and percentages (%).

	COVID-19 Prevention and Control		
	No (*n* = 43)	Yes (*n* = 42)	*p*-Value
Neutropenia (grade 3 or 4)	43 (100)	40 (95.2)	0.241
Neutropenia (grade 4)	40 (93.0%)	38 (90.5)	0.713
FN	12 (27.9%)	4 (9.5%)	0.050
Admission	17 (39.5)	9 (21.4%)	0.099
Hospitalization duration (days)	2.0 ± 3.8	0.7 ± 1.5	0.041
Outpatient costs (USD)	227.0 ± 166.7	253.8 ± 58.7	0.325
Inpatient costs (USD)	137.7 ± 186.4	26.4 ± 60.1	0.001
Total cost (USD)	364.7 ± 271.6	279.6 ± 42.6	0.049
Dose reduction or delay	7 (16.3)	3 (7.1)	0.313
Anemia	3 (7.0)	1 (2.4)	0.616
Thrombocytopenia	8 (18.6)	6 (14.3)	0.771
Liver function abnormality	5 (11.6%)	7 (16.7)	0.549
Renal failure	1 (2.3)	0 (0)	1.000
Transfusion	4 (9.3)	2 (4.8)	0.676
Infection	5 (11.6)	4 (9.5)	1.000
Death	2 (4.7)	0	0.494

**Table 3 jcm-11-07053-t003:** Odds of febrile neutropenia according to logistic regression analysis.

Variable	OR	95% Confidence Interval	*p*-Value
Age (years)			
<60	1		
≥60	1.700	0.540–5.347	0.364
Menopause			
Premenopausal	1		
Menopausal	0.947	0.307–2.917	0.924
BMI (kg/m^2^)			
<25	1		
≥25	0.688	0.248–1.903	0.471
IPC guidance			
Yes	1		
No	3.677	1.078–12.543	0.038
Comorbidities			
No	1		
Yes	4.026	1.081–14.990	0.038
Histologic type			
IDC	1		
ILC	2.231	0.188–26.496	0.525
Others	0.991	0.191–5.142	0.992
T stage			
T1	1		
T2	0.524	0.159–1.726	0.288
T3–T4	7.333	0.595–90.332	0.120
N stage			
0	1		
1	2.545	0.503–12–891	0.259
2	2.154	0.336–13.804	0.418
3	1.400	0.103–19.012	0.800
Estrogen receptor			
Negative	1		
Positive	1.268	0.426–3.775	0.669
Progesterone receptor			
Negative	1		
Positive	0.610	0.195–1.905	0.395
HER2			
Negative	1		
Positive	1.301	0.400–4.234	0.662

BMI, body mass index; BSA, body surface area; DM, diabetes mellitus; HER2, human epidermal growth factor receptor 2; IPC, infection prevention and control; IDC, infiltrating ductal carcinoma; ILC, infiltrating lobular carcinoma.

**Table 4 jcm-11-07053-t004:** Odds of febrile neutropenia according to multivariate logistic regression analysis.

Variable	OR	95% Confidence Interval	*p*-Value
IPC guidance			
Yes	1		0.023
No	4.668	1.238–17.602	
Comorbidities			
No	1		0.021
Yes	5.554	1.294–23.843	

IPC, infection prevention and control.

## Data Availability

The data presented in this study are available on request from the corresponding author.

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
