# Peer review of "COVID-19 Prevention Guidance and the Incidence of Febrile Neutropenia in Patients with Breast Cancer Receiving TAC Chemotherapy with Prophylactic Pegfilgrastim"

_jcm, 2022, doi:10.3390/jcm11237053_

Round 1

Reviewer 1 Report

1. I strongly advise researchers to include medical oncologists at their institution to review their work. Ciprofloxiacin and growth factor does not cause acute leukemias (line 161, page 6). There's no black box warnings regarding that. Doxorubicin and cyclophosphamide, however, can do that and have black box warnings about that. Furthermore, neutropenic fever in the breast cancer population is typically managed with brief hospitalization and sometimes even oral antibiotics, they are typically low risk for sepsis and significant complications. Prophylaxis with fluoroquinolones is also not routinely indicated for TAC. see: https://academic.oup.com/cid/article/52/4/e56/382256

2. Infection prevention/control clearly did NOT decrease neutropenic fever in your results. Your discussion needs to explain that.

3. Seems like the IPC was implemented at different times of the year based on the pandemic, could the hospitals tried to expedite cancer patients out of the hospital so they don't get COVID or perhaps to accommodate COVID patients? Breast cancer patients with neutropenic fever can be treated with oral antibiotics if they are stable, maybe that's why their hospitalization was shorter. Or maybe staff are just more careful during COVID? Please discuss these issues.

4. Please discuss the time aspects between IPC and non-IPC group, perhaps in relationship to the COVID pandemic in South Korea.

Author Response

Thanks for the advice. I have corrected all the parts you mentioned and described the content in the answer. In addition, it is displayed as a memo in the manuscript.

This manuscript has been re-edited to ensure language and grammar accuracy and is error free in these aspects. The edit was performed by professional editors at Editage, a division of Cactus Communications.

Point 1: I strongly advise researchers to include medical oncologists at their institution to review their work. Ciprofloxiacin and growth factor does not cause acute leukemias (line 161, page 6). There's no black box warnings regarding that. Doxorubicin and cyclophosphamide, however, can do that and have black box warnings about that. Furthermore, neutropenic fever in the breast cancer population is typically managed with brief hospitalization and sometimes even oral antibiotics, they are typically low risk for sepsis and significant complications. Prophylaxis with fluoroquinolones is also not routinely indicated for TAC. see: https://academic.oup.com/cid/article/52/4/e56/382256

Response 1: Thanks for the advice. A medical oncologist was included in the review. We have deleted and modified the contents as you pointed out about the drugs used for prevention and their side effects. (page 6, line 163-166).

Point 2: Infection prevention/control clearly did NOT decrease neutropenic fever in your results. Your discussion needs to explain that.

Response 2: Thanks for the advice. I have corrected the part you mentioned and added the relevant content back to the discussion (Page 4 line 125-126 & 6 line 177-180).

Point 3: Seems like the IPC was implemented at different times of the year based on the pandemic, could the hospitals tried to expedite cancer patients out of the hospital so they don't get COVID or perhaps to accommodate COVID patients? Breast cancer patients with neutropenic fever can be treated with oral antibiotics if they are stable, maybe that's why their hospitalization was shorter. Or maybe staff are just more careful during COVID? Please discuss these issues.

Response 3: Thanks for the advice. Pandemic situations have the potential to affect a patient's treatment policy. However, the study was conducted in a state where there was almost no outbreak, and it is thought that it would not have had much effect by trying to treat it according to the existing protocol, and this content was added to the discussion (page 7 line 204-216).

Point 4: Please discuss the time aspects between IPC and non-IPC group, perhaps in relationship to the COVID pandemic in South Korea.

Response 4: The time aspect of IPC and non-IPC group was described in the discussion(page 7 line 206-213). Thank you

Reviewer 2 Report

1. The Introduction part did not provide enough background information about the study. Actually, some descriptions in the Discussion part should be moved to the Introduction. And meanwhile, the Discussion part should focus on the findings of the current study. Please rewrite these two parts if possible. Thank you.

2. The Estrogen Receptor of patients with or without IPC shows different. Is this factor contribute to the conclusion too?

3. The single-center retrospective observation is not a good method to conduct the current study. It is difficult to draw the conclusion the authors would like to propose.

Author Response

Thanks for the advice. I have corrected all the parts you mentioned and described the content in the answer. In addition, it is displayed as a memo in the manuscript.

This manuscript has been re-edited to ensure language and grammar accuracy and is error free in these aspects. The edit was performed by professional editors at Editage, a division of Cactus Communications.

Point 1: The Introduction part did not provide enough background information about the study. Actually, some descriptions in the Discussion part should be moved to the Introduction. And meanwhile, the Discussion part should focus on the findings of the current study. Please rewrite these two parts if possible. Thank you.

Response 1: Thanks for the advice. Some of the content in the discussion has been moved to the introduction(page 1 line 38-45). In addition, the findings of this study were further described in the discussion(page 6 line 173-180).

Point 2: The Estrogen Receptor of patients with or without IPC shows different. Is this factor contribute to the conclusion too?

Response 2: Estrogen receptors did not affect the conclusions of this study. Regarding the part you mentioned, the contents of the research results have been added(page 4 line 137-139 & page 6 line 171-173) to the text and related references are also attached.

Point 3: The single-center retrospective observation is not a good method to conduct the current study. It is difficult to draw the conclusion the authors would like to propose.

Response 3: Thanks for the advice that made my paper a better one. I've added what you said to the discussion(page 7 line 201-204 & 216-219).

Round 2

Reviewer 1 Report

Amazing edits and response. The only suggestion I have is in Line 165 in Discussion: However, in some patients, FN can be a severe adverse effect that requires intravenous broad-spectrum antibiotics.... etc. 

Author Response

Dear reviewer.

This paper received additional English revision through the editing service. I added a sentence as you advised (page 6, line 173). Your advice made my article better. Thank you

Reviewer 2 Report

The authors made improvements to the original version. The introduction part still could not provide enough background information. Also, the study lacks major scientific insights.

Author Response

Dear reviewer.

This paper received additional English revision through the editing service.

I added sentences as you advised in introduction(page 2, line 48-54) and discussion (page 6, line 179-182, page 7, line 200-201).

Your advice made my paper better. Thank you